# *Clostridioides difficile* Infections in Children: What Is the Optimal Laboratory Diagnostic Method?

**DOI:** 10.3390/microorganisms12091785

**Published:** 2024-08-28

**Authors:** Mohammed Suleiman, Patrick Tang, Omar Imam, Princess Morales, Diyna Altrmanini, Jill C. Roberts, Andrés Pérez-López

**Affiliations:** 1Department of Pathology, Sidra Medicine, Doha P.O. Box 26999, Qatar; ptang@sidra.org (P.T.); pmorales@sidra.org (P.M.); daltrmanini@sidra.org (D.A.); aperezlopez@sidra.org (A.P.-L.); 2Department of Pathology and Laboratory Medicine, Weill Cornell Medicine in Qatar, Doha P.O. Box 24144, Qatar; 3Department of Infectious Diseases, Sidra Medicine, Doha P.O. Box 26999, Qatar; oimam@sidra.org; 4Center for Global Health and Infectious Disease Research, University of South Florida, Tampa, FL 33612, USA; jcrobert@usf.edu

**Keywords:** *Clostridioides difficile*, diagnostic methods, pediatric, children

## Abstract

The diagnosis of *Clostridioides difficile* infection (CDI) in the pediatric population is complicated by the high prevalence of asymptomatic colonization, particularly in infants. Many laboratory diagnostic methods are available, but there continues to be controversy over the optimal laboratory testing approach to diagnose CDI in children. We evaluated commonly used *C. difficile* diagnostic commercial tests in our pediatric hospital population at Sidra Medicine in Doha, Qatar. Between June and December 2023, 374 consecutive stool samples from pediatric patients aged 0–18 years old were tested using: Techlab *C. diff* Quik Chek Complete, Cepheid GeneXpert *C. difficile*, QIAstat-Dx Gastrointestinal Panel, and culture using CHROMagar *C. difficile*. The results of these tests as standalone methods or in four different testing algorithms were compared to a composite reference method on the basis of turnaround time, ease of use, cost, and performance characteristics including specificity, sensitivity, negative predictive value, and positive predictive value. Our study showed variability in test performance of the different available assays in diagnosing CDI. In our population, a testing algorithm starting with Cepheid GeneXpert *C. difficile* PCR assay or QIAstat-Dx Gastrointestinal panel as a screening test followed by toxin immunoassay for positive samples using the Techlab *C. diff* Quik Chek Complete kit showed the best performance (100% specificity and 100% positive predictive value) when combined with clinical review of the patient to assess risk factors for CDI, clinical presentation, and alternative causes of diarrhea.

## 1. Introduction

*Clostridioides difficile* is a spore-forming, anaerobic gram-positive bacterium that can cause serious gastrointestinal disease in humans [1,2]. *C. difficile* infection (CDI) is a common healthcare-associated infection and the leading cause of antibiotic-associated diarrhea worldwide in all age groups [3]. Also, it has been associated with community-acquired infections and recurrent CDIs in the pediatric population [4]. In the United States, CDI has been associated with 15,000 to 30,000 deaths in 2011 and over $4.8 billion in excess acute care inpatient costs in 2008 [5,6,7]. Although *C. difficile* is a public health concern, there is a lack of consensus on which laboratory method is the best to use, as there is no single laboratory test that is rapid, cost-effective, highly sensitive, and specific [5,8].

The diagnosis of CDI in children is complicated by the high prevalence of asymptomatic carriage of both non-toxigenic and toxigenic strains in the pediatric population [5]. A systematic review from 2021 showed asymptomatic *C. difficile* colonization ranging from 13.6% to 53.8% among formula-fed and breastfed infants [9]. Although a majority of these infants carried non-toxigenic strains, asymptomatic colonization with toxigenic strains was still significant, ranging from 4.8% to 5.8% [9]. The rate of colonization also decreases with age [5,10], but it remains elevated during the second year of life [5,11]. By 2 to 3 years of age, the rate of asymptomatic colonization in children drops to 2–3%, which is similar to the rate observed in healthy adults [5]. The asymptomatic carriage of *C. difficile* in younger children makes the clinical interpretation of laboratory tests more challenging.

There are a variety of laboratory test methods available to diagnose CDI using stool samples. Toxigenic culture (TC) detects the growth of the actual organism, but it is labor-intensive, time-consuming, and requires a pre-treatment step and anaerobic incubation [2,5,12,13]. TC is considered one of the gold standard methods to detect *C. difficile* and is often used in epidemiological studies. However, the performance of TC varies significantly based on the method and technique used [2,5,12,13]. Cell culture cytotoxicity assay (CTA) is another reference method that detects toxin directly in stool by observing the cytopathic effects. Similar to TC, it is very complex, labor-intensive, and time-consuming [2,5,13]. Enzyme immunoassays (EIAs) for the detection of *C. difficile* toxin A and B became commercially available in the late 1980s and replaced the previous reference methods above for routine clinical testing [5]. Although these EIAs are easy to use and easily implemented in clinical laboratories, the overall poor performance (low sensitivity and specificity) of these assays sparked the development of other methods such as detection of glutamate dehydrogenase (GDH) and detection of toxin genes using nucleic acid amplification tests (NAAT) [2,5,13]. GDH immunoassays are easy to use and detect a common antigen present in all isolates of *C. difficile* (toxigenic and nontoxigenic strains), which makes it a highly sensitive test [2,5,13]. However, GDH immunoassays have lower clinical specificity and must be confirmed with another (usually toxin EIA or NAAT) test [2,5,13]. Molecular testing for *C. difficile* DNA in stool samples started in the early 1990s, and the first NAAT was cleared by the US Food and Drug Administration (FDA) in 2009 [5]. Currently, many FDA-approved molecular assays (singleplex PCR or within multiplex panels) are available for the detection of the genes encoding for *C. difficile* toxins (*tcdA* and *tcdB*) [2,5,13]. These NAAT assays are more sensitive for *C. difficile* detection than GDH and toxin EIAs. However, these NAAT assays lack specificity to confirm CDI and are unable to differentiate symptomatic infection from colonization, especially in the pediatric population [2,5,13].

The European Society of Clinical Microbiology and Infectious Diseases (ESCMID) recommends using a multistep algorithm starting with a sensitive screening test such as NAAT or GDH EIA followed, in the case of a positive result, a confirmatory test (CTA) is recommended to confirm the presence of toxins in stool specimens [2]. Alternatively, a combined GDH and toxin EIA may be employed for screening, followed by NAAT confirmation for GDH-positive and toxin-negative results [2]. The Infectious Diseases Society of America (IDSA) and the Society for Healthcare Epidemiology of America (SHEA) recommend using algorithms similar to those recommended by ESCMID; however, they also recommend using NAATs alone as a first-step method when the test is performed on stool samples from children ≥ 2 years of age with prolonged or worsening diarrhea; the patients have risk factors for CDI such as recent antibiotic exposure, underlying immunocompromising conditions, or inflammatory bowel disease (IBD); and the patients have not received any laxatives in the past 48 h [2,5]. A recent clinical review showed that IBD is an independent risk factor for CDI, affecting both the pediatric and elderly populations [14]. In addition, IBD patients are at increased risk of having recurrent CDIs, as well as increased complications resulting from CDI [5]. Management and treatment of children with CDI is similar to adults and includes the use of metronidazole, vancomycin, or fecal microbiota transplantation in patients with recurrent CDIs [5].

Most *C. difficile* studies focus on adults, and there are a limited number of studies focusing on the pediatric population [5,13]. In addition, many of these studies are limited by the absence or inappropriate use of a gold-standard diagnostic method (TC or CTA) and a lack of knowledge about the prevalence rate of CDI in the studied population [5]. It remains unclear if one method should be used alone or within a multistep algorithm and which method is preferred in the pediatric population, as many patients may be colonized with *C. difficile* without having the actual infection [5]. The purpose of this study is to assess and compare the performance of different diagnostic methods and algorithms for the detection and diagnosis of CDI in pediatric patients.

## 2. Materials and Methods

### 2.1. Specimen Collection and Study Design

Consecutive, fresh stool samples, liquid or soft, collected from patients aged 0 to 18 years old and submitted to our microbiology laboratory at Sidra Medicine for routine *C. difficile* PCR testing or gastrointestinal PCR panel between June and December 2023 were included in this study. Located in the State of Qatar, Sidra Medicine is a tertiary care pediatric hospital serving as the main pediatric subspecialty referral center for the country. The hospital handles 28,000 pediatric admissions, 115,000 emergency department visits, and 190,000 outpatient appointments on an annual basis. The study included all pediatric samples collected from outpatient units, inpatient units, and emergency services. Samples were collected in sterile containers, and diagnostic testing was immediately performed upon receipt in our laboratory. Following our standard sample rejection criteria, formed stools and duplicate samples within seven days from the same patient were excluded. Repeat samples from previous PCR-positive patients within the 30 days were also excluded. Clinical data was collected, including patient sex, patient age, patient location, previous or current antibiotic exposure, previous history of CDI, and clinical symptoms such as fever and diarrhea.

### 2.2. C. difficile Laboratory Diagnostic Tests

All consecutive samples were tested prospectively for the presence of toxins A and B and GDH using the *C. diff* Quik Chek Complete kit immunoassay (CDQ; Techlab, Blacksburg, VA, USA), detection of toxin A and B genes using the QIAstat-Dx Gastrointestinal Panel 2 PCR test (QGP; QIAGEN, Hilden, Germany), and detection of toxin B gene using the Xpert *C. difficile* PCR test (XCD; Cepheid, Sunnyvale, CA, USA). All tests were performed, and results were interpreted according to the manufacturer’s instructions.

In addition, all samples were cultured on CHROMagar *C. difficile* (CAC; CHROMagar, Paris, France). The stool samples were streaked onto the plates using plastic loops and were incubated anaerobically at 35–37 °C for 24 h. The plates were read at 24 h using a UV lamp at 365 nm, and any suspicious (colorless and fluorescent under UV lamp) colonies were identified using the Bruker Matrix-assisted laser desorption/ionization-time of flight mass spectrometry (MALDI-TOF MS) system (Bruker Daltonics, Bremen, Germany) according to the manufacturer’s instructions. The isolates were directly smeared onto the disposable target plate, then layered with a small drop of the α-cyano-4-hydroxycinnamic acid matrix solution and air dried. The prepared target plate was then loaded into the Bruker MALDI-TOF MS system for analysis.

### 2.3. Statistical Analysis

Specificity, sensitivity, negative predictive value (PPV), and positive predictive value (NPV) were calculated with 95% confidence intervals for each method alone (GDH or Toxin of the CDQ kit, or XCD or QGP or CAC). In addition, specificity, sensitivity, NPV, and PPV were also calculated with 95% confidence intervals for the four different algorithms combining multiple methods together. Algorithm 1 is a two-step algorithm with the CDQ kit as the first test, followed by performing XCD PCR on GDH (+)/Toxin (−) samples. Algorithm 2 is a two-step algorithm with the CDQ kit as the first test, followed by performing QCP PCR on GDH (+)/Toxin (−) samples. Algorithm 3 is a two-step algorithm with XCD PCR as the first test, followed by the toxin test from the CDQ kit for PCR positive samples. Algorithm 4 is a two-step algorithm with QGP PCR as the first test, followed by the toxin test from the CDQ kit for PCR positive samples.

The performance of the five methods and the four algorithms was assessed by comparing test results to a composite gold standard. In the analysis, the sample was considered TP if it was positive by all five testing methods. The sample was considered true negative (TN) if it was negative using all five testing methods. If the sample was positive by one to four methods out of five methods, a clinical review of the patient history was conducted to determine if the patient has a true CDI. The patient was considered TP if they had worsening or prolonged diarrhea (more than 3 loose stools), no alternative diagnosis, and had one of the risk factors for CDI. The risk factors included: antibiotic exposure in the last six weeks, immunosuppressants, oncology diagnosis, solid organ transplant, or IBD.

## 3. Results

### 3.1. Samples and Demographics

A total of 374 consecutive stool samples meeting the exclusion and inclusion criteria of our study were tested using all five testing methods. The median age of patients in our study was 7 years old, and the interquartile range was 8.75. Fifty-five (14.7%) samples were collected from patients less than 2 years old, and 319 (85.3%) samples were from patients aged 2–18 years old. Two hundred (53.5%) samples originated from male patients and 174 (46.5%) from female patients. Overall, 170 (45.5%) samples were collected from the hospital inpatient units, while 69 (18.4%) were from the emergency department, and 135 (36.1%) were from outpatient clinics. Thirty (8%) samples were collected from oncology patients, and 43 (11.5%) samples were collected from IBD patients. Fifty-nine (15.7%) patients had a previous history of CDI, and 164 (43.8%) patients had previous antibiotic exposure.

### 3.2. Performance Characteristics Analysis

A comparison of turnaround time (TAT), ease of use, and cost of each method and/or algorithm is detailed in Table 1. The sensitivity, specificity, PPV, and NPV of individual methods and algorithms are detailed in Table 2, Figure 1 and Figure 3, for patients aged 2–18 years old. Table 3, Figure 2 and Figure 4, show the data for patients < 2 years old. Ninety samples (24.1%) tested positive by at least one of the five testing methods. Using the composite reference method, 16 samples were considered as TP, with an overall prevalence of 4.2% in all age groups. In children aged 2–18 years old, the prevalence was 14/319 (4.4%), and in children < 2 years old, the prevalence was 2/55 (3.6%).

**Figure 1 microorganisms-12-01785-f001:**
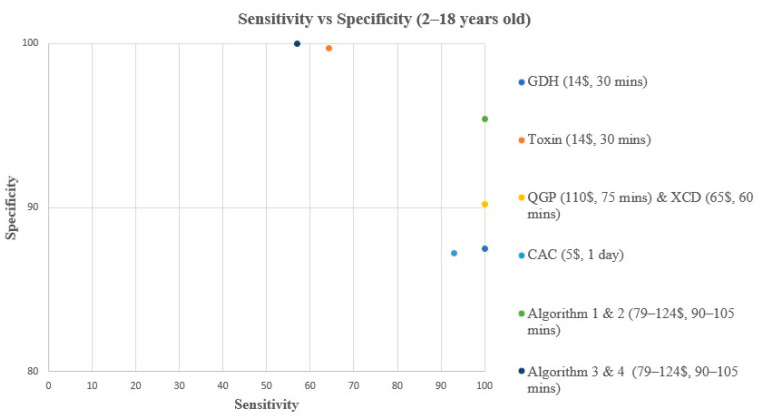
Sensitivity vs. Specificity comparison for all methods & algorithms in 2–18 years old. CAC, CHROMagar *C. difficile*; CDQ, *C. diff* Quik Chek Complete kit immunoassay; GDH, Glutamate dehydrogenase; QGP, QIAstat-Dx Gastrointestinal Panel 2 PCR test; XCD, Xpert *C. difficile* PCR test. Algorithm 1, CDQ arbitrated by XCD; Algorithm 2, CDQ arbitrated by QIAstat-Dx Multiplex PCR; Algorithm 3, XCD followed by toxin A/B from the CDQ kit; Algorithm 4, QGP followed by toxin A/B from the CDQ kit.

**Figure 2 microorganisms-12-01785-f002:**
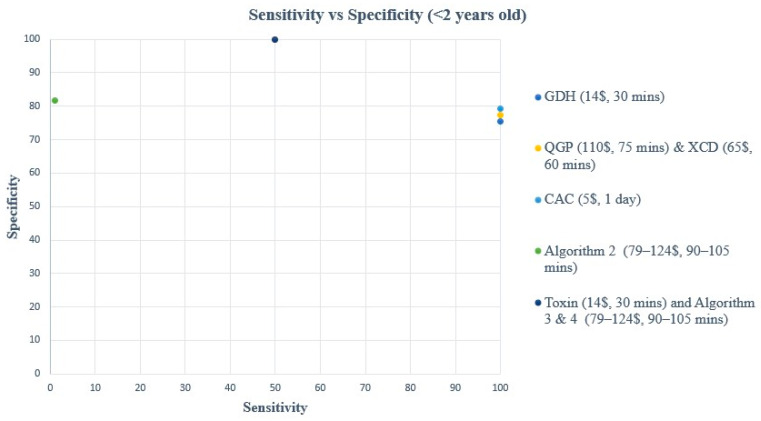
Sensitivity vs. Specificity comparison for all methods & algorithms in <2 years old. CAC, CHROMagar *C. difficile*; CDQ, *C. diff* Quik Chek Complete kit immunoassay; GDH, Glutamate dehydrogenase; QGP, QIAstat-Dx Gastrointestinal Panel 2 PCR test; XCD, Xpert *C. difficile* PCR test. Algorithm 2, CDQ arbitrated by QIAstat-Dx Multiplex PCR.

**Figure 3 microorganisms-12-01785-f003:**
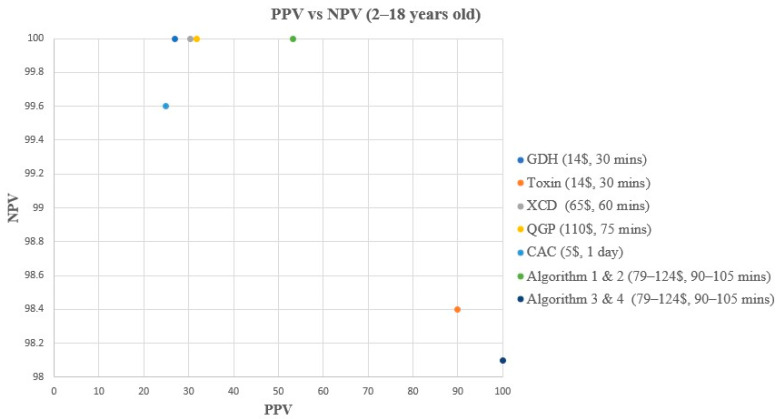
PPV vs. NPV comparison for all methods & algorithms in 2–18 years old. CAC, CHROMagar *C. difficile*; CDQ, *C. diff* Quik Chek Complete kit immunoassay; GDH, Glutamate dehydrogenase; NPV, Negative predictive value; PPV, Positive predictive value; QGP, QIAstat-Dx Gastrointestinal Panel 2 PCR test; XCD, Xpert *C. difficile* PCR test. Algorithm 1, CDQ arbitrated by XCD; Algorithm 2, CDQ arbitrated by QIAstat-Dx Multiplex PCR; Algorithm 3, XCD followed by toxin A/B from the CDQ kit; Algorithm 4, QGP followed by toxin A/B from the CDQ kit.

**Figure 4 microorganisms-12-01785-f004:**
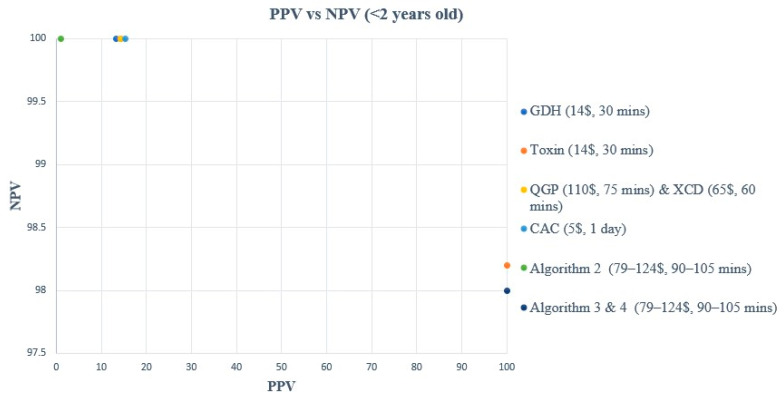
PPV vs. NPV comparison for all methods & algorithms in <2 years old. CAC, CHROMagar *C. difficile*; CDQ, *C. diff* Quik Chek Complete kit immunoassay; GDH, Glutamate dehydrogenase; NPV, Negative predictive value; PPV, Positive predictive value; QGP, QIAstat-Dx Gastrointestinal Panel 2 PCR test; XCD, Xpert *C. difficile* PCR test. Algorithm 2, CDQ arbitrated by QIAstat-Dx Multiplex PCR; Algorithm 3, XCD followed by toxin A/B from the CDQ kit; Algorithm 4, QGP followed by toxin A/B from the CDQ kit.

No hypervirulent NAP1 strains were detected in any of the samples using the XCD method. Using the QIAstat-Dx Gastroenteritis multiplex PCR Panel, 139 samples out of the 374 samples tested were positive for pathogens other than *C. difficile*, including 119 samples with other bacterial pathogens, 33 samples with viral pathogens, and 9 samples with a parasite. 63 samples out of the 139 were positive for multiple pathogens, including 20 that were also positive for *C. difficile*.

In children aged 2–18 years old, all methods and algorithms demonstrated excellent specificity and NPV ranging from 87% to 100% (Table 2, Figure 1 and Figure 3). Algorithms 3 and 4 starting with XCD or QGP PCR followed by the toxin method are the most specific (100%), with the highest PPV (100%), and 0 false positive (FP) samples (Table 2, Figure 1 and Figure 3). The PPV was low for CAC, GDH, XCD, QGP methods, and algorithms 1 and 2, ranging from 25% to 53.3% (Table 2, Figure 1 and Figure 3). Sensitivity varied greatly between methods and algorithms, with the toxin method and algorithms 3 and 4 showing the lowest sensitivity (57.1–64.3%), while the GDH, XCD PCR, QGP PCR, and algorithms 1 and 2 showed the highest sensitivity (100%) (Table 2, Figure 1 and Figure 3).

In children < 2 years old, all methods and algorithms showed excellent NPV (98.1 to 100%), while a great variation was observed in terms of sensitivity, specificity, and PPV (Table 3, Figure 2 and Figure 4). The toxin method and algorithms 3 and 4 show low sensitivity of 50% but high PPV and specificity (100%) (Table 3, Figure 2 and Figure 4). On the other hand, the GDH, CAC, XCD, QGP, and algorithm 2 show excellent sensitivity of 100% but low PPV ranging from 13.3 to 20% (Table 3, Figure 2 and Figure 4). The sensitivity and PPV could not be calculated for algorithm 1 due to the absence of TP samples using this algorithm.

## 4. Discussion

Although toxin immunoassay methods are highly specific, cheap, and relatively easy to perform, the toxin detection showed low sensitivity in our study, which concurs with previous studies in the pediatric population [15,16,17,18]. The lower sensitivity of these toxin tests is consistent with younger children being more likely to carry non-toxigenic strains and lower rates of NAP1 strains in children [18,19]. In fact, no NAP1 strains of *C. difficile* were detected in our pediatric patient population. Another contributing factor may be that the policies against the testing of asymptomatically colonized children may not have been strictly enforced in our institution. The results of our study support the recommendations from IDSA-SHEA that standalone stool toxin tests are inadequate for the diagnosis of CDI [5]. GDH, the other component of the CDQ kit, showed better sensitivity than the toxin component, but it resulted in 51 FP samples, suggesting that GDH is not helpful for differentiating colonization from infection and cannot distinguish toxigenic from non-toxigenic strains of *C. difficile*. Due to the very low prevalence in our population (4.2%), the PPV for GDH was lower than estimates from three previous studies that showed PPV ranging from 50% to 67% in the context of higher prevalence (16% to 24%) [15,18,20]. Another study showed an even higher PPV (94%), but the *C. difficile* prevalence in their study population was even higher (36% of samples were positive, and 43% of patients had CDI) [16].

Seventeen percent of laboratories in the United States use a combined GDH and toxin immunoassay test followed by NAAT for GDH positive and toxin negative results as recommended by the IDSA-SHEA [5,21]. This algorithm (1 and 2 in our study) is quick, easy to perform, and highly specific. However, in our pediatric population, this algorithm (1 and 2) showed low PPV (48–53.3%) in patients 2–18 years old. This is mainly due to the many FPs from the GDH component in this algorithm. Our finding of lower sensitivity of these algorithms is consistent with previous studies in the pediatric population [15,16,18]. The reason for this lower sensitivity remains unclear, but it has been suggested that this could be due to the low incidence of hypervirulent strains of *C. difficile,* such as NAP1 strains, in the pediatric population [15,18,22,23].

Many of the laboratories in the United States (40%) are using NAAT as a standalone test to diagnose CDI, which is also recommended by IDSA-SHEA [5,21]. NAAT methods are more costly and have longer TAT than toxin and GDH, but they are more sensitive and specific. Both NAAT methods in our study (XCD and QGP) showed very similar results. They both produced excellent results in terms of sensitivity (100%), NPV (100%), and specificity (89.5–90.1% in patients 2–18 years old and 77.4–81.8% in patients < 2 years old). However, they showed poor PPV (30.4–31.8% in patients 2–18 years old and 14.3% in patients < 2 years old). These results suggest that NAAT assays might not be able to differentiate colonization from true CDI in the pediatric population, especially children younger than 2 years old, and reliance on NAAT assays as standalone tests to diagnose *C. difficile* might result in overdiagnosis and overtreatment [2,5,13]. Although several studies in the literature evaluated NAAT methods for the diagnosis of CDI and showed excellent results in terms of test performance [15,16,17,18,20,24,25,26,27,28,29,30,31,32], most of these studies did not correlate clinical data with the test results to diagnose CDI. Due to the superior sensitivity of NAAT in comparison to traditional methods, many labs have switched to NAAT assays, which has resulted in a significant increase in the reported rates of CDI in health care facilities [17,20,27,33]. This has raised the concern that many asymptomatic carriers of toxigenic *C. difficile* may have been incorrectly diagnosed with CDI.

Five previous studies evaluated the performance of syndromic panels for the detection of *C. difficile* [24,25,26,29,34]. These studies showed excellent sensitivity (81–100%) and excellent specificity (85–100%), but they all used another NAAT method as a reference method. In contrast, our study is the first to evaluate the QGP in a pediatric population and the first to compare a multiplex PCR syndromic panel against multiple comparator methods. In addition to the high sensitivity in detecting *C. difficile*, the QGP panel was helpful in finding an alternative diagnosis in 37% (139 of 374) of the samples and resulted in two fewer FP results in comparison to the XCD. Also, the QGP panel diagnosed CDI in 3 patients where *C. difficile* testing was not requested, which was important for ensuring that those patients were diagnosed correctly and treated for CDI. This is similar to a previous study where *C. difficile* was detected in 8% of samples where *C. difficile* testing was not ordered [35]. This study suggested that specific physician ordering practices might miss a diagnosis of CDI, and the use of multiplex panels would increase the CDI detection rate as well as the overall detection of other important gastrointestinal pathogens [35].

Both testing methods discussed above (NAAT only, GDH/Toxin arbitrated by NAAT) are weak recommendations by the IDSA-SHEA with weak evidence in the literature, especially in the pediatric population. To improve the laboratory algorithm to diagnose CDI, we evaluated several new methods and algorithms which have not been well studied previously. CAC is the cheapest, easy to perform, but has the longest TAT of 24 h in comparison to all other methods. The test performance of CAC was very similar to that of GDH. CAC showed excellent sensitivity (92.9% in patients 2–18 years old and 100% in patients < 2 years old) and good specificity (87.2% in patients 2–18 years old and 79.3% in patients < 2 years old). On the other hand, it resulted in 50 FP results and low PPV (25% in patients 2–18 years old and 15.4% in patients < 2 years old).

The reverse algorithm (3 and 4) with the NAAT test first followed by toxin testing showed promising results. Only one previous pediatric study assessed this algorithm, but it focused on the clinical aspects rather than diagnostic accuracy and concluded that this algorithm failed to differentiate symptomatic CDI from asymptomatic colonization [36]. The same study also concluded that pediatric patients who were toxin positive are more likely to have more severe symptoms compared to patients who were toxin negative [36]. Our study is the first to assess both the diagnostic accuracy and clinical significance of this algorithm in a pediatric population. These algorithms showed excellent specificity (100%), excellent PPV (100%), and excellent NPV (98%). However, these algorithms had lower sensitivity compared with algorithms 1 and 2 (57.1% in patients 2–18 years old and 50% in patients < 2 years old). Therefore, our findings suggest that patients with positive toxin results using this algorithm can be treated with confidence. Conversely, patients with negative toxin results would need a review of their clinical symptoms and risk factors to determine whether they are colonized or have active CDI.

This study had several limitations; although we tested a high number of prospective samples (n = 374), our pediatric population had a relatively low prevalence of *C. difficile*. As the traditional gold standard methods of TC or CTA were not available in our laboratory, we used a composite gold standard including the correlation between the test results with the clinical presentation of the patient to assess for CDI. The CTA and TC assays have different targets in that the CTA detects free toxin in samples while TC detects vegetative cells or spores, and a positive result indicates that the patient is infectious [30]. As opposed to the traditional gold standard assays, NAAT methods target the toxin genes [37]. Many recent *C. difficile* studies in the pediatric population do not use the traditional TC or CTA as their gold standard method but instead use PCR or did not use any reference methods at all [24,25,26,29,33,34,37,38,39,40,41,42,43,44]. Finally, our institution did not have strict rejection criteria regarding patient pre-test requirements, which would increase the pre-test probability of CDI. Testing criteria recommended by the IDSA-SHEA that were used in our composite analysis include testing patients with prolonged or worsening diarrhea (≥3 times in 24 h) and relevant exposures or risk factors [5]. Also, the IDSA-SHEA does not recommend *C. difficile* testing as a test for cure [5].

Several areas would benefit from further research. Our study included pediatric patients with a wide range in age and immune status. Studying the test performance in these different sub-populations could be valuable in future studies. The use of PCR cycle thresholds to predict toxin load has been evaluated in several recent studies, and it could be useful to assess the performance of this approach in our pediatric population [39,40,44,45].

In conclusion, the diagnosis of CDI in children remains a challenge due to the high rate of colonization in children and lack of consensus regarding gold standards. Our study showed variability in test performance of the different available assays in diagnosing CDI. An algorithm starting with sensitive XCD or QGP as a screening test followed by a specific toxin assay for positive samples using the CDQ showed the best performance in our population when combined with clinical review of the patient symptoms, alternative diagnoses, and risk factors for CDI. To further improve the diagnostic efficiency of *C. difficile* testing, health care facilities should implement strict criteria to ensure testing is only performed on pediatric patients with a higher pre-test probability of having CDI.

## Figures and Tables

**Table 1 microorganisms-12-01785-t001:** Comparison of methods and algorithms based on TAT, ease of use, and cost.

Method	TAT	Ease of Use	Cost
GDH	30 min	Simple, manual procedure	11–20 USD
Toxin	30 min	Simple, manual procedure	11–20 USD
XCD	60 min	Simple, automated procedure	21–100 USD
QGP	75 min	Simple, automated procedure	>100 USD
CAC	24 h	Simple, manual procedure	<10 USD
Algorithm 1	90 min	Simple, manual, and automated procedure	21–100 USD
Algorithm 2	105 min	Simple, manual, and automated procedure	>100 USD
Algorithm 3	90 min	Simple, manual, and automated procedure	21–100 USD
Algorithm 4	105 min	Simple, manual, and automated procedure	>100 USD

CAC, CHROMagar *C. difficile*; CDQ, *C. diff* Quik Chek Complete kit immunoassay; GDH, Glutamate dehydrogenase; QGP, QIAstat-Dx Gastrointestinal Panel 2 PCR test; USD, United States Dollars; XCD, Xpert *C. difficile* PCR test. Algorithm 1, CDQ arbitrated by XCD; Algorithm 2, CDQ arbitrated by QIAstat-Dx Multiplex PCR; Algorithm 3, XCD followed by toxin A/B from the CDQ kit; Algorithm 4, QGP followed by toxin A/B from the CDQ kit.

**Table 2 microorganisms-12-01785-t002:** Performance characteristics of individual methods and algorithms (2–18 years old).

Method	Reference	TP	TN	FP	FN	Sensitivity (%)	Specificity (%)	PPV (%)	NPV (%)
GDH	Composite	14	267	38	0	100 (78.5–100)	87.5 (83.3–90.8)	26.9 (16.8–40.3)	100 (98.6–100)
Toxin	Composite	9	304	1	5	64.3 (38.8–83.7)	99.7 (98.2–100)	90 (59.6–98.2)	98.4 (96.3–99.3)
XCD	Composite	14	273	32	0	100 (78.5–100)	89.5 (85.6–92.5)	30.4 (19–44.8)	100 (98.6–100)
QGP	Composite	14	275	30	0	100 (78.5–100)	90.2 (86.3–93)	31.8 (20–45.6)	100 (98.6–100)
CAC	Composite	13	266	39	1	92.9 (68.5–98.7)	87.2 (83–90.5)	25 (15.2–38.2)	99.6 (97.9–99.9)
Algorithm 1	Composite	16	289	14	0	100 (80.6–100)	95.4 (92.4–97.2)	53.3 (36.1–69.8)	100 (98.7–100)
Algorithm 2	Composite	14	290	15	0	100 (78.5–100)	95.1 (92.1–97)	48.3 (31.4–65.6)	100 (98.7–100)
Algorithm 3	Composite	8	305	0	6	57.1 (32.6–78.6)	100 (98.8–100)	100 (67.6–100)	98.1 (95.9–99.1)
Algorithm 4	Composite	8	305	0	6	57.1 (32.6–78.6)	100 (98.8–100)	100 (67.6–100)	98.1 (95.9–99.1)

CAC, CHROMagar *C. difficile*; CDQ, *C. diff* Quik Chek Complete kit immunoassay; FN, false negative; FP, false positive; GDH, glutamate dehydrogenase; NPV, negative predictive value; PPV, positive predictive value; QGP, QIAstat-Dx gastrointestinal Panel 2 PCR test; TN, true negative; TP, true positive; XCD, Xpert *C. difficile* PCR test. Algorithm 1, CDQ arbitrated by XCD; Algorithm 2, CDQ arbitrated by QIAstat-Dx Multiplex PCR; Algorithm 3, XCD followed by toxin A/B from the CDQ kit; Algorithm 4, QGP followed by toxin A/B from the CDQ kit.

**Table 3 microorganisms-12-01785-t003:** Performance characteristics of individual methods and algorithms (<2 years old).

Method	Reference	TP	TN	FP	FN	Sensitivity (%)	Specificity (%)	PPV (%)	NPV (%)
GDH	Composite	2	40	13	0	100 (34.2–100)	75.5 (62.4–85.1)	13.3 (3.7–37.9)	100 (91.2–100)
Toxin	Composite	1	53	0	1	50 (9.5–90.6)	100 (93.2–100)	100 (20.7–100)	98.2 (90.2–99.7)
XCD	Composite	2	41	12	0	100 (34.2–100)	77.4 (64.5–86.6)	14.3 (4–39.9)	100 (91.4–100)
QGP	Composite	2	41	12	0	100 (34.2–100)	77.4 (64.5–86.6)	14.3 (4–39.9)	100 (91.4–100)
CAC	Composite	2	42	11	0	100 (34.2–100)	79.3 (66.5–88)	15.4 (4.3–42.2)	100 (91.4–100)
Algorithm 1	Composite	0	45	10	0	NA	81.8 (69.7–89.8)	NA	100 (92.1–100)
Algorithm 2	Composite	2	45	8	0	100 (34.2–100)	84.9 (73–92.1)	20 (5.7–51)	100 (92.1–100)
Algorithm 3	Composite	1	53	0	1	50 (9.5–90.6)	100 (93.2–100)	100 (20.7–100)	98.2 (90.2–99.7)
Algorithm 4	Composite	1	53	0	1	50 (9.5–90.6)	100 (93.2–100)	100 (20.7–100)	98.2 (90.2–99.7)

CAC, CHROMagar *C. difficile*; CDQ, *C. diff* Quik Chek Complete kit immunoassay; FN, False negative; FP, False positive; GDH, Glutamate dehydrogenase; NPV, Negative predictive value; PPV, Positive predictive value; QGP, QIAstat-Dx Gastrointestinal Panel 2 PCR test; TN, True negative; TP, True positive; XCD, Xpert *C. difficile* PCR test. Algorithm 1, CDQ arbitrated by XCD; Algorithm 2, CDQ arbitrated by QIAstat-Dx Multiplex PCR; Algorithm 3, XCD followed by toxin A/B from the CDQ kit; Algorithm 4, QGP followed by toxin A/B from the CDQ kit.

## Data Availability

The original contributions presented in the study are included in the article, further inquiries can be directed to the corresponding author.

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
