# Peer review of "Clostridioides difficile Infections in Children: What Is the Optimal Laboratory Diagnostic Method?"

_microorganisms, 2024, doi:10.3390/microorganisms12091785_

Round 1

Reviewer 1 Report

Comments and Suggestions for Authors

I have read with interest the manuscript submitted by Suleiman et al, since CDI represents a global concern.

I have a few comments to be addressed in order to improve the quality of the manuscript:

- the plagiarism identified through iThenticate should be properly addressed. It is neither acceptable to copy paragraphs from previously published articles.

- the introduction should focus more on the particularities of CDI in the pediatric population

- the results section should include some figures

- some more information about the patients could be included 

- what was the positivity rate? how many samples were tested in total?

- the authors should describe why they separated the patients into the two groups and why choosing the age of 2

- statistical analysis should be performed to compare sensitivity and specificity among the studied groups

- references are not cited uniformly. I suggest using only [].

- the discussion section should not include referrals to tables from the results section.

- some more information regarding the difference between colonization with toxigenic strains and infection in pediatric patients, should be included

- the authors could also briefly mention if there are any management differences between the pediatric and adult populations.

Best regards,

Comments on the Quality of English Language

minor editing required

Reviewer 2 Report

Comments and Suggestions for Authors

I reviewed this interesting study with great interest, which compared the diagnostic performance of various approaches to diagnosing C. difficile infection in the pediatric setting. Here are some considerations:

A significant limitation of the study is the absence of statistical analyses, such as a ROC analysis or statistical comparisons between the sensitivity, specificity, PPV, and NPV of the different techniques. In other words, the techniques appear to be compared only through descriptive statistics on diagnostic performance parameters. It would have been valuable to statistically compare the AUROCs of the different techniques.

Other minor considerations:

1. In addition to the median, I would also provide the interquartile range for each continuous variable.

2. Given that 43 (11.5%) of your samples were detected in patients with IBD, I would clarify in the introduction, with slightly more detail, the importance of this infectious risk factor in the pathogenesis of IBD (https://pubmed.ncbi.nlm.nih.gov/38676422/), as well as the fact that IBD patients may be more susceptible to developing this infection even when antibiotics are justified, such as for eradicating H. pylori infection (https://doi.org/10.3390/diseases12080179).

Reviewer 3 Report

Comments and Suggestions for Authors

The paper presents an important topic of the steps for making the diagnosis of Clostridioides difficile infection in pediatric population.

Abstract: Consider being specific about the obtained results – i.e. include numerical values.

Introduction: The second sentence of the first paragraph that cites references 3 and 4 – there are more recent published reports. Also, seems the last access date was more than a year ago. Reference No. 4 actually addresses an important aspect of CDI – community acquired, as well as recurrent infections. However, the reference is cited in a sentence that is about something else – check and find and use references that refer to what is stated.

Introduction: The third sentence needs to state the year for which data on costs are provided.

Introduction: First paragraph should provide a clear picture of the epidemiology of CDI in children, with provided burden estimates to illustrate the magnitude of the problem, especially since it is not enough to state that “C. difficile is clearly still a public health concern”.

Introduction: Give the rates and locations for the mentioned high prevalence of asymptomatic carriage in children.

A clear overview of both the European and American guidelines i.e. diagnostic algorithms is provided.

Materials and methods: Provide details on the population – how many patients does the hospital treat on an annual basis, how many are there with diarrhea, were all consecutive stool samples included, how was the studied period decided on – that the number of collected samples was enough to allow testing of different diagnostic algorithms? Details on all of this must be added.

Results are well presented.

Discussion section provides a comprehensive interpretation of results, compared to other original studies, and provides a good attempt to give explanations for obtained results.

Round 2

Reviewer 1 Report

Comments and Suggestions for Authors

I appreciate the author's efforts in addressing my comments. The quality of the manuscript has improved. My only minor comments would be:

- the newly inserted paragraphs should undergo some English editing

- further information about the distinguishment between colonization with toxigenic strains and infection in the pediatric population should be inserted.

- each figure should contain a legend describing the abbreviations used. Also, I suggest adding the figures in the results section, not at the end of the manuscript for greater visibility.

- the reference list is not edited according to the mdpi pattern, especially the newly inserted references.

Best regards,

Comments on the Quality of English Language

- the newly inserted paragraphs should undergo some English editing

Reviewer 2 Report

Comments and Suggestions for Authors

revisions done

Author Response

Thank you for your review 

Reviewer 3 Report

Comments and Suggestions for Authors

I would like to thank the Authors for carefully revising their manuscript and answering my comments.

With the changes made in the revision, the Introduction section now reads better and provides a good snapshot of the magnitude and thus significance of C. difficile infections, representing a significant public health issue still to this moment in time.

The section Methods underwent a comprehensive revision, and the requested information was added, especially in the 2.1 subsection. 

Comments on the Quality of English Language

The English language of the manuscript is good.

Author Response

Thank you for your review and comments